# Associations of Heavy Metals with Metabolic Syndrome and Anthropometric Indices

**DOI:** 10.3390/nu12092666

**Published:** 2020-09-01

**Authors:** Wei-Lun Wen, Chih-Wen Wang, Da-Wei Wu, Szu-Chia Chen, Chih-Hsing Hung, Chao-Hung Kuo

**Affiliations:** 1Department of Internal Medicine, Kaohsiung Municipal Siaogang Hospital, Kaohsiung Medical University, Kaohsiung 812, Taiwan; stevenwen760829@gmail.com (W.-L.W.); chinwin.wang@gmail.com (C.-W.W.); u8900030@yahoo.com.tw (D.-W.W.); kjh88kmu@gmail.com (C.-H.K.); 2Division of Endocrinology and Metabolism, Department of Internal Medicine, Kaohsiung Medical University Hospital, Kaohsiung Medical University, Kaohsiung 807, Taiwan; 3Division of Hepatobiliary, Department of Internal Medicine, Kaohsiung Medical University Hospital, Kaohsiung Medical University, Kaohsiung 807, Taiwan; 4Division of Pulmonary and Critical Care Medicine, Department of Internal Medicine, Kaohsiung Medical University Hospital, Kaohsiung Medical University, Kaohsiung 807, Taiwan; 5Division of Nephrology, Department of Internal Medicine, Kaohsiung Medical University Hospital, Kaohsiung Medical University, Kaohsiung 807, Taiwan; 6Research Center for Environmental Medicine, Kaohsiung Medical University, Kaohsiung 807, Taiwan; pedhung@gmail.com; 7Department of Pediatrics, Kaohsiung Medical University Hospital, Kaohsiung Medical University, Kaohsiung 807, Taiwan; 8Department of Pediatrics, Kaohsiung Municipal Siaogang Hospital, Kaohsiung Medical University, Kaohsiung 812, Taiwan; 9Division of Gastroenterology, Department of Internal Medicine, Kaohsiung Medical University Hospital, Kaohsiung Medical University, Kaohsiung 807, Taiwan

**Keywords:** heavy metals, metabolic syndrome, anthropometric indices

## Abstract

Previous studies have revealed associations between heavy metals and extensive health problems. However, the association between heavy metals and metabolic problems remains poorly defined. This study aims to investigate relationships between heavy metals and metabolic syndrome (MetS), lipid accumulation product (LAP), visceral adiposity index (VAI), and anthropometric indices, including body roundness index (BRI), conicity index (CI), body adiposity index (BAI), and abdominal volume index (AVI). We conducted a health survey of people living in southern Taiwan. Six heavy metals were measured: lead (Pb) in blood and nickel (Ni), chromium (Cr), manganese (Mn), arsenic (As), and copper (Cu) in urine. A total of 2444 participants (976 men and 1468 women) were enrolled. MetS was defined according to the Adult Treatment Panel III for Asians. Multivariable analysis showed that participants with high urine Ni (log per 1 μg/L; odds ratio (OR): 1.193; 95% confidence interval (CI): 1.019 to 1.397; *p* = 0.028) and high urine Cu (log per 1 μg/dL; OR: 3.317; 95% CI: 2.254 to 4.883; *p* < 0.001) concentrations were significantly associated with MetS. There was a significant trend of a stepwise increase in blood Pb and urine Ni, As, and Cu according to the number of MetS components (from 0 to 5, a linear *p* ≤ 0.002 for trend). For the determinants of indices, urine Cu was positively correlated with LAP, BRI, CI, and VAI; blood Pb was positively correlated with BRI, BAI, and AVI; urine Ni was positively correlated with LAP. High urine Cu and urine Ni levels were significantly associated with MetS, and there was a significant trend for stepwise increases in blood Pb and urine Ni, As, and Cu, accompanied by an increasing number of MetS components. Furthermore, several indices were positively correlated with urine Cu, urine Ni, and blood Pb.

## 1. Introduction

Awareness of environmental protection has increased in recent years, and several international environmental agreements have been adopted to restrict the harmful impact of all kinds of pollution, including noise, air, marine, oil, organic compounds, inorganic compounds, radioactive waste, and heavy metals, on the environment and public health [1]. Among these pollutants, heavy metals may be the most challenging as they do not easily degrade [2]. Furthermore, current remediation technologies are complex and expensive [3], and this hinders the removal of these toxic agents from the environment to prevent accumulation in the bioactive food chain as industrialization and urbanization progress [4]. Heavy metals are generally defined as metals or metalloids with a density >4–5 g/cm^3^ [5]. Sources of contamination can be classified as being anthropogenic or natural, with anthropogenic sources including industrialization, mining, and the agricultural use of metal-containing compounds [6]. Natural sources of contamination are also known as pedogenesis [7]. Heavy metals enter the human body through ingestion or contact with contaminated food, water, soil, and air [2]. Previous studies have reported associations between heavy metals and extensive health problems, including neurodegenerative, cardiovascular, obstructive lung, renal, hematological, bone and teeth, acute gastrointestinal diseases, and cancers [8,9,10,11,12,13,14]. However, the association between heavy metals and metabolic problems remains poorly defined.

Metabolic syndrome (MetS) is defined as a pathologic condition characterized by abdominal obesity, insulin resistance, hypertension, and dyslipidemia [15]. The prevalence of MetS in Taiwan is approximately 16%, according to a nationwide cross-sectional population-based survey conducted in the early 2000s [16]. Several indices have been developed as surrogate markers of central obesity and insulin resistance and have been shown to be highly correlated with the diagnosis of MetS and the risk of further developing diabetes or atherosclerotic cardiovascular diseases [17,18,19]. The potential pathways of these central obesity-representing indices associated with MetS could be related to visceral fat, increasing production of nonesterified fatty acids (NEFAs), inflammatory cytokines, prothrombotic factors such as plasminogen activator inhibitor (PAI-1), and the activation of renin–angiotensin–aldosterone (RAAS) system, but decreasing production of an anti-inflammatory and antiatherogenic adipokine, adiponectin [20,21]. These indices include lipid accumulation product (LAP), visceral adiposity index (VAI), and anthropometric indices, including body roundness index (BRI), conicity index (CI), body adiposity index (BAI), and abdominal volume index (AVI), all of which can easily be calculated and quantified using factors such as waist circumference, hip circumference, body mass index (BMI), body height (BH), body weight (BW), triglycerides (TGs), and high-density lipoprotein (HDL) cholesterol [17].

The underlying mechanism of heavy metals toxicity is complex, but a large part of it can be attributed to cell apoptosis or carcinogenesis after the exhaustion of native cellular antioxidants, which fail to balance heavy-metal-induced reactive oxygen species (ROS) production anymore [22]. This oxidative stress might also decrease the insulin gene promoter activity and mRNA expression in pancreatic islet cells; moreover, some of the toxic metals are implicated in disrupting glucose uptake and alter the related molecular mechanism in glucose regulation directly [23]. All of the above could be dedicated to the relationship between heavy metals and metabolic syndrome. Previous studies have reported associations between MetS and different heavy metals. A higher prevalence of MetS has been associated with higher blood levels of lead (Pb) [24] or higher urinary levels of copper (Cu) and zinc (Zn) [25]. Furthermore, a study disclosed significantly higher serum levels of manganese (Mn) and chromium (Cr) in obese aged men, which might indirectly intensify MetS; lower serum levels of magnesium (Mg) directly increased the risk of MetS in the same population [26]. There were also some studies that focused on the association between heavy metals and body composition but more inconsistently. Chromium (Cr) supplementation was associated with mild bodyweight loss and a decrease in body fat percentage in obese or overweight participants [27]. Serum Zn concentrations were positively associated with both abdominal obesity and total body fat in men but not in women [28]. Neither blood Pb concentrations nor urine arsenic (As) concentrations were associated with the fat mass percentage in reproductive-age women [29].

In this study, we collected more than 2000 health survey results from people living in southern Taiwan to investigate the relationships between blood Pb, urine nickel (Ni), urine Cr, urine Mn, urine As, and urine Cu levels and MetS and different anthropometric indices.

## 2. Materials and Methods

### 2.1. Subject Recruitment

We conducted a health survey of the general population in southern Taiwan from June 2016 to September 2018. Participants who knew about the survey from advertisements and were willing to attend the study were included. Anthropometric variables (weight, height, systolic blood pressure (SBP) and diastolic blood pressure (DPB), WC, and HC) were measured, and an experienced physician performed physical examinations and recorded clinical histories (including hypertension and diabetes).

### 2.2. Collection of Demographic, Medical, and Laboratory Data

The following variables were recorded at baseline: demographics (age and sex), medical history (DM and hypertension), examination findings (SBP and DBP), and laboratory data (fasting glucose, TGs, total, HDL- and LDL-cholesterol, hemoglobin, uric acid, and estimated glomerular filtration rate (eGFR)). EGFR was calculated using the Chronic Kidney Disease Epidemiology Collaboration equation (CKD-EPI eGFR) [30]. BMI was calculated as weight/height squared (kg/m^2^).

### 2.3. Measurement of Blood and Urine Heavy Metals

Six heavy metals were measured: Pb in blood and Ni, Cr, Mn, As, and Cu in urine. The concentrations of these heavy metals were analyzed using graphite furnace atomic absorption spectrometry (ICP-MS, NexION 300 Series, Perkin Elmer). Details of the instrumental analysis have been reported by the National Institute of Environmental Research.

### 2.4. Definition of MetS

The National Cholesterol Education Program Adult Treatment Panel (NCEP-ATP) III guidelines [31] and the modified criteria for Asians [32] were used to evaluate MetS, which was defined as having three of the following five abnormalities: (1) high blood pressure (SBP ≥ 130 mmHg, DBP ≥ 85 mmHg) or a diagnosis/treatment for hypertension; (2) hyperglycemia (fasting whole-blood glucose concentration ≥110 mg/dL or DM); (3) low concentration of HDL-cholesterol (<40 mg/dL in men and <50 mg/dL in women); (4) hypertriglyceridemia (TG concentration ≥150 mg/dL); (5) abdominal obesity (WC > 90 cm for men and >80 cm for women).

### 2.5. Indices

LAP was calculated as:

LAP = $\left(\;WC_{\left(cm\right)}-65\right)\times \;TG_{\left(mmol/L\right)}$ in males, and 

LAP = $\left(\;WC_{\left(cm\right)}-58\right)\times \;TG_{\left(mmol/L\right)}$ in females [33]. 

BRI was calculated as: 

BRI = $364.2-365.5\;\times \;\sqrt{1-{\left(\frac{\frac{WC_{\left(m\right)}}{2\pi }}{0.5\;\times \;BH_{\left(m\right)}}\right)}^{2}}$ [34]. 

CI was calculated using the Valdez equation based on BW, BH and WC as: 

CI = $\frac{WC_{\left(m\right)}}{0.109\;\times \;\sqrt{\frac{BW_{\left(kg\right)}}{BH_{\left(m\right)}}}}$ [35]. 

VAI score was calculated as described previously [36] using the following sex-specific equations (with TG levels in mmol/L and HDL-cholesterol levels in mmol/L): 

VAI = $\left(\frac{WC_{\left(cm\right)}}{39.68\;+\left(1.88\;\times \;BMI\right)}\right)\times \left(\frac{TG_{\left(mmol/L\right)}}{1.03}\right)\times \left(\frac{1.31}{HDL_{\left(mmol/L\right)}}\right)$ in males, and 

VAI = $\left(\frac{WC_{\left(cm\right)}}{36.58\;+\left(1.89\;\times \;BMI\right)}\right)\times \left(\frac{TG_{\left(mmol/L\right)}}{0.81}\right)\times \left(\frac{1.52}{HDL_{\left(mmol/L\right)}}\right)$ in females.

BAI was calculated according to the method of Bergman and colleagues as:

BAI = $\frac{\;Hip\;circumference_{\left(cm\right)\;}}{BH_{\left(m\right)}{}^{3/2}}\;-18$ [37]. 

AVI was calculated as AVI = $\frac{2\;\times \;{\left(\;WC_{\left(cm\right)}\;\right)}^{2}+0.7\;\times \;{\left(\;WC_{\left(cm\right)}-HC{\;}_{\left(cm\right)}\right)}^{2}}{1000}$ [38]. 

### 2.6. Ethics Statement

The study protocol was approved by the Institutional Review Board of Kaohsiung Medical University Hospital (number: KMUHIRB-G(II)-20190011). All participants provided informed consent before study enrollment.

### 2.7. Statistical Analysis

Statistical analysis was performed using SPSS version 19.0 for Windows (SPSS Inc. Chicago, IL, USA). Data were expressed as percentages, mean ± standard deviation, or median (25th–75th percentile) for TGs and the heavy metals. Between-group differences were analyzed using the chi-square test for categorical variables and independent *t*-test for continuous variables. Multiple comparisons among study groups were performed using a one-way analysis of variance followed by Bonferroni-adjusted posthoc analysis. Multivariable logistic regression analysis was used to identify associations between heavy metals and MetS. Multivariable linear regression analysis was used to identify associations between heavy metals and LAP, VAI, and anthropometric indices. For all heavy metal measurements (in blood and urine), the natural logarithm was used. A *p*-value of less than 0.05 was considered to indicate a statistically significant difference.

## 3. Results

The mean age of the 2444 participants (976 males and 1468 females) was 55.1 ± 13.2 years. The overall prevalence rate of MetS was 33.8%. A comparison of the clinical characteristics among the participants, with and without MetS, is shown in Table 1. Compared to the participants without MetS, those with MetS were older and had higher prevalence rates of DM and hypertension and higher BMI, WC, HC, SBP, DBP, fasting glucose, hemoglobin, uric acid, and TGs. In addition, they had lower HDL-cholesterol and eGFR than the participants without MetS. Regarding heavy metals, participants with MetS had higher levels of blood Pb, urine Ni, urine As, and urine Cu. In addition, the participants with MetS had higher anthropometric indices, including LAP, BRI, CI, VAI, BAI, and AVI.

### 3.1. Determinants of MetS

Table 2 shows the determinants of MetS in the study participants. After adjusting for each heavy metal, age, sex, total cholesterol, LDL-cholesterol, hemoglobin, eGFR and uric acid, the participants with high concentrations of urine Ni (log per 1 µg/L; odds ratio (OR): 1.193; 95% confidence interval (CI): 1.019 to 1.397; *p* = 0.028) and high urine Cu (log per 1 µg/dL; OR: 3.317; 95% CI: 2.254 to 4.883; *p* < 0.001) were significantly associated with MetS.

Table 3 shows heavy metal values according to the sum of MetS components in the participants. The concentration of blood Pb and urine Ni, As, and Cu increased with the number of MetS components (from 0 to 5, a linear *p* ≤ 0.002 for trend). The multiple-testing-adjusted correction *p*-values were computed by stepdown Bonferroni (threshold of *p* = 0.0083; p0/N1, p0 = 0.05, N1 = 6 risk factors, see Table 3).

### 3.2. Determinants of Each Index

Table 4 shows the determinants of each index in the participants after multivariable linear regression analysis, as follows:

#### 3.2.1. LAP

After multivariable adjustment, high urine Ni (log per 1 µg/L; unstandardized coefficient β: 2.418; 95% CI: 0.288 to 4.548; *p* = 0.026), and high urine Cu (log per 1 µg/dL; unstandardized coefficient β: 9.508; 95% CI: 4.406 to 14.609; *p* < 0.001) were significantly associated with high LAP.

#### 3.2.2. BRI

After multivariable adjustment, high blood Pb (log per 1 µg/dL; unstandardized coefficient β: 0.190; 95% CI: 0.019 to 0.362; *p* = 0.030), and high urine Cu (log per 1 µg/dL; unstandardized coefficient β: 0.223; 95% CI: 0.038 to 0.407; *p* = 0.018) were significantly associated with high BRI.

#### 3.2.3. CI

After multivariable adjustment, high urine Cu (log per 1 µg/dL; unstandardized coefficient β: 0.014; 95% CI: 0.002 to 0.027; *p* = 0.023) was significantly associated with high CI.

#### 3.2.4. VAI

After multivariable adjustment, high urine Cu (log per 1 µg/dL; unstandardized coefficient β: 0.898; 95% CI: 0.149 to 1.646; *p* = 0.023) was significantly associated with high VAI.

#### 3.2.5. BAI

After multivariable adjustment, high blood Pb (log per 1 µg/dL; unstandardized coefficient β: 1.093; 95% CI: 0.241 to 1.944; *p* = 0.012) was significantly associated with high BAI.

#### 3.2.6. AVI

After multivariable adjustment, high blood Pb (log per 1 µg/dL; unstandardized coefficient β: 0.726; 95% CI: 0.120 to 1.332; *p* = 0.019) was significantly associated with high AVI.

## 4. Discussion

In this study, we found that urine Cu and Ni were associated with MetS and that an increase in the number of MetS components was associated with an increase in blood Pb and urine Ni, As, and Cu. For the determinants of indices, urine Cu was positively correlated with LAP, BRI, CI, and VAI, blood Pb was positively correlated with BRI, BAI, and AVI, and urine Ni was positively correlated with LAP.

Cu is an essential trace element that is stored in the liver, and it is mostly excreted in bile. Several key copper-containing enzymes are involved in copper–iron interactions and central nervous system function, so anemia and myeloneuropathy may be manifestations of Cu deficiency [39,40]. Humans mainly absorb Cu from food, and Cu toxicity is usually caused by accident, occupational exposure, environmental contamination, or inborn errors of Cu metabolism. Excess Cu in the body affects health, mainly by increasing the formation of ROS [41]. Previous studies have investigated the associations between Cu and neurodegenerative and liver diseases. In the present study, a high urine Cu concentration was associated with MetS, which is consistent with a recent study by Ma et al. (urine copper concentration quartile: <5.4975 versus >12.6323 μg/L) [25]. Furthermore, we found a trend of a stepwise increase in urine Cu and the number MetS components and also a positive correlation with LAP, BRI, CI, and VAI, which has not been reported before. In terms of a possible mechanism, Ma et al. also revealed that urine Cu increased monotonically with the marker of inflammation C-reactive protein [25]. This could explain the association between Cu and MetS, as C-reactive protein has also been associated with MetS [42]. In the present study, we did not have the data of C-reactive protein. We could not compare the data with other studies.

Exposure to Ni in daily life can occur through contact with stainless steel kitchen utensils or electronic devices contained nickel alloys, inhaling tobacco or lead-free fuels, and ingesting nickel-containing foods [43,44]. No human enzyme or cofactor contains Ni, and Ni affects health by inducing the production of ROS and activating an immune response to cause allergy. Previous research has focused on carcinogenicity and Ni allergy [45]. Yang et al. reported a relatively higher prevalence of MetS in 35,000 Chinese workers who were exposed to Ni compared to the general population (13.9% versus 13.2%), but they did not measure the workers’ urine or serum Ni concentrations [46]. In addition, the reference values for urine Ni in healthy adults is 1–3 μg/L [47]. In the present study, similar to Cu, we found that a high urine Ni concentration was associated with MetS, increasing the number of MetS components, and positively correlated with LAP, which has not been reported before. In terms of a possible mechanism, an animal study exposing mice to nickel chloride or particulate matter of 10 μm (PM_10_), collected from Saudi Arabia, where Ni concentrations are relatively high compared to other locations, possibly due to the burning of oil which produces Ni, found that the two groups of mice displayed similar dysregulation of metabolic genes [48]. In addition, a high Ni concentration in the diet has been suggested to affect the composition of microbiota [45] and cause an increase in the Enterobacter species [49], and this has been related to obesity or metabolic dysfunction [50].

Lead is another common pollutant, and the most likely exposure routes in the general population are ingestion of contaminated food and drinking water, possibly through water transmitted by lead pipes or food contained in improperly glazed pottery dishes. Furthermore, vehicle emission of lead in the air is also one of the sources of Pb exposure in areas still using leaded gasoline [51]. Similar to Ni, Pb has not been found to play any physiological role in the human body; the effects of Pb poisoning include generating ROS, damaging cell membrane integrity by interfering with vitamin D synthesis, and interfering with DNA transcription [52]. Pb toxicity has been associated with neurological, cardiovascular, renal, hematopoietic, immune, and reproductive system disorders [53]. One nationwide, population-based, cross-sectional study in Korea suggested that a higher prevalence of MetS was associated with higher blood Pb levels (log-transformed lead concentration quartile: 3.07–19.43 versus 0.42–1.73 μg/dL) [24]. In the present study, we did not find a significant association between blood Pb and MetS after adjustments in multivariable logistic regression analysis, probably due to the fact that Pb levels in our study population were far lower than the formerly mentioned study or even the upper limit of the Centers for Disease Control and Prevention (CDC) reference value, 5 μg/dL [51]. In the present study, we found a significant trend of a stepwise increase in blood Pb according to the number of MetS components, and blood Pb was positively correlated with BRI, BAI, and AVI. As mentioned, these anthropometric indices are all surrogate markers of MetS, and a recent study revealed that all of these indices, except for AVI, could accurately predict MetS [18]. In summary, blood Pb is correlated with central obesity and the number of MetS components. The possible mechanism may be through the overproduction of ROS, irritation of the absorption surface, interactions with specific proteins to impair DNA repair, or the induction of cell apoptosis [6].

Chronic arsenic exposure threatens the health of more than 100 million people worldwide, where drinking water sources are groundwater with higher concentrations of this toxic metalloid, including certain areas of Taiwan [54]. Besides blackfoot disease or cancers (skin, bladder, lung), which are increasingly found in high-dose long-term exposure, increased risk of MetS has also been reported. Wang et al. have reported an increased prevalence of MetS from the 2nd tertile of hair As concentrations in a population at the central part of Taiwan, but the urine As concentration data was not available [55]. In the present study, though we did not find a direct relationship between urine As levels and metabolic syndrome, we did find the urine As trend was accompanied by an increasing number of MetS components. The discrepancy might be attributed to no measurement of the composition of total urine As, which could be divided into inorganic arsenic (iAs), dimethylarsinic acid (DMA), and monomethylarsonic acid (MMA). The latter two organic As ratios represent individualized arsenic methylation capability, which has been suspected to be a more important factor of metabolic risk than just total urine As concentration in no matter high or low to moderately As-exposed population (urine total As: 6.5 to ~43.46 μg/L) [56,57].

There was no association between urine Cr or Mn and MetS or any anthropometric index in our study. Chromium (III) or trivalent chromium is an essential trace element that plays a role in carbohydrate and lipid metabolism in the human body and it is normally present in blood and urine, with the mean level of 0.10–0.16 and 0.22 μg/L, respectively [58]. As mentioned before, Cr supplements could decrease BW and improve anthropometric indices in the obese but not an already-DM population [27]. Chronic exposure to Cr was found to decrease MetS development in the United States, but not in the Korean population [59,60]. The reason why our study did not reveal a similar relationship between concentrations and MetS or anthropometric indices may be due to generally lower Cr exposure levels, similar to the Korean study. Mn is also an essential trace element, and the mean urinary manganese concentration in the general population is about 1.19 μg/L [61]. Unlike Cr, its deficiency is rarely reported, and it is still inconclusive whether a low Mn diet in humans is related to MetS or not [62,63]. In contrast, Mn poisoning and its neurotoxicity are well documented, possibly due to the neurons’ long lifespan and high energy demand. Although Mn also accumulates in the pancreas, no definite evidence of its toxicity related to MetS has been reported, similar to our study [64].

MetS per se and its compartments have been related to an important public health threat and economic burden by the further development of diabetes or cardiovascular events [65,66], and early diagnosis, follow-up, and long-term management are recommended [67]. Therefore, the government might not only make policy to prevent future heavy metal pollution but also continuously monitor the exposers of specific heavy metals, which we have revealed in this article to have possible associations with MetS.

There are several limitations to this study. First, this is a cross-sectional study, so we could not define the causal relationship between these heavy metals and MetS. Second, a single laboratory test may not reflect Pb exposure, as it can accumulate in bones and the kidneys and be slowly released into blood or urine [52]. Third, due to restrictions with the study design, we could only perform spot urine tests, which are less precise to evaluate Cu intoxication than 24-h urine [68]. Fourth, because only participants that were willing to attend the study were included, this makes it considerably more difficult to interpret standard errors and confidence intervals. In addition, the source of these heavy metals cannot be confirmed. Finally, we checked total As but not iAs, DMA, MMA, and their ratios in urine due to equipment restrictions, and this may have underestimated the contribution of As to MetS.

In conclusion, we found that high urine Cu and Ni levels were significantly associated with MetS, and there was also a significant trend for stepwise increases in urine Cu, Ni, and As and blood Pb accompanied by an increasing number of MetS components. Further, several anthropometric indices, including LAP, BRI, CI, VAI, BAI, and AVI, were positively correlated with urine Cu, urine Ni, and blood Pb.

## Figures and Tables

**Table 1 nutrients-12-02666-t001:** Comparison of clinical characteristics among participants, with and without MetS.

Characteristics	All (*n* = 2444)	Without MetS (*n* = 1618)	With MetS (*n* = 826)	*p*
Age (year)	55.1 ± 13.2	52.9 ± 13.0	59.4 ± 12.6	<0.001
Male gender (%)	39.9	39.2	41.3	0.331
DM (%)	10.5	4.2	22.8	<0.001
Hypertension (%)	25.3	16.9	41.8	<0.001
BMI (kg/m2)	25.0 ± 4.0	23.7 ± 3.4	27.5 ± 3.8	<0.001
Waist circumference (cm)	83.6 ± 10.6	80.0 ± 9.7	90.7 ± 8.8	<0.001
Hip circumference (cm)	96.5 ± 8.0	94.7 ± 7.4	100.0 ± 7.9	<0.001
SBP (mmHg)	132.1 ± 19.8	126.8 ± 18.4	142.3 ± 18.3	<0.001
DBP (mmHg)	77.5 ± 11.7	75.4 ± 11.0	81.6 ± 11.9	<0.001
Laboratory parameters				
Fasting glucose (mg/dL)	99.9 ± 27.4	91.9 ± 16.5	115.5 ± 36.3	<0.001
Triglyceride (mg/dL)	105.0 (73.0–150.0)	87.0 (65.0–115.0)	161.0 (118.0–215.3)	<0.001
Total cholesterol (mg/dL)	199.6 ± 37.5	199.7 ± 36.1	199.6 ± 40.1	0.966
HDL-cholesterol (mg/dL)	53.0 ± 13.6	57.2 ± 13.3	44.7 ± 10.1	<0.001
LDL-cholesterol (mg/dL)	119.2 ± 34.0	119.0 ± 32.7	119.5 ± 36.4	0.771
Hemoglobin (g/dL)	14.0 ± 1.6	13.9 ± 1.6	14.1 ± 1.7	0.003
eGFR (mL/min/1.73 m2)	89.1 ± 16.3	91.8 ± 14.7	83.8 ± 18.1	<0.001
Uric acid (mg/dL)	5.7 ± 1.6	5.5 ± 1.5	6.2 ± 1.6	<0.001
Heavy metals				
Blood				
Pb (µg/dL)	1.6 (1.0–2.2)	1.5 (1.0–2.2)	1.6 (1.1–2.3)	0.002
Urine				
Ni (µg/L)	2.4 (1.5–3.7)	2.4 (1.5–3.7)	2.5 (1.6–3.8)	0.005
Cr (µg/L)	0.1 (0.1–0.1)	0.1 (0.1–0.1)	0.1 (0.1–0.1)	0.953
Mn (µg/L)	1.7 (0.9–3.0)	1.7 (0.9–2.9)	1.7 (0.9–3.0)	0.324
As (µg/L)	78.9 (45.6–142.0)	74.9 (42.9–131.9)	87.8 (50.7–158.3)	<0.001
Cu (µg/dL)	1.5 (1.0–2.0)	1.4 (1.0–1.8)	1.6 (1.2–2.1)	<0.001
LAP	35.1 ± 34.5	21.6 ± 15.4	61.7 ± 44.7	<0.001
BRI	3.9 ± 1.3	3.4 ± 1.1	4.8 ± 1.2	<0.001
Anthropometric indices				
CI	1.2 ± 0.1	1.2 ± 0.1	1.3 ± 0.1	<0.001
VAI	4.3 ± 4.8	2.8 ± 1.9	7.2 ± 6.9	<0.001
BAI	29.8 ± 5.0	28.8 ± 4.6	31.8 ± 5.2	<0.001
AVI	13.9 ± 4.3	13.3 ± 3.1	16.6 ± 3.2	<0.001

Abbreviations: MetS, metabolic syndrome; DM, diabetes mellitus; BMI, body mass index; SBP, systolic blood pressure; DBP, diastolic blood pressure; HDL, high-density lipoprotein; LDL, low-density lipoprotein; eGFR, estimated glomerular filtration rate; Pb, lead; Ni, nickel; Cr, chromium; Mn, manganese; As, arsenic; Cu, copper; LAP, lipid accumulation product; BRI, body roundness index; CI, conicity index; VAI, visceral adiposity index; BAI, body adiposity index; AVI, abdominal volume index.

**Table 2 nutrients-12-02666-t002:** Association of heavy metals with MetS using multivariable logistic regression analysis.

Heavy Metals	Multivariable	
	OR (95% CI)	*p*
Blood		
Pb (log per 1 µg/dL)	0.857 (0.613–1.199)	0.368
Urine		
Ni (log per 1 µg/L)	1.193 (1.019–1.397)	0.028
Cr (log per 1 µg/L)	0.845 (0.487–1.466)	0.549
Mn (log per 1 µg/L)	1.035 (0.873–1.227)	0.691
As (log per 1 µg/L)	0.933 (0.717–1.215)	0.608
Cu (log per 1 µg/dL)	3.317 (2.254–4.883)	<0.001

Values expressed as odds ratio (OR) and 95% confidence interval (CI). Abbreviations are the same as in Table 1. Covariates in the multivariable model included age, sex, total cholesterol, LDL-cholesterol, hemoglobin, eGFR, and uric acid.

**Table 3 nutrients-12-02666-t003:** Heavy metal values according to the sum of components of MetS in study participants.

Heavy Metals (Log-Transformation)	0 (*n* = 475)	1 (*n* = 574)	2 (*n* = 569)	3 (*n* = 464)	4 (*n* = 266)	5 (*n* = 96)	*p* for Trend
Blood							
Pb (µg/dL)	0.12 ± 0.01	0.15 ± 0.01	0.18 ± 0.01 *	0.19 ± 0.01 *	0.20 ± 0.02 *	0.15 ± 0.03	<0.001
Urine							
Ni (µg/L)	0.20 ± 0.03	0.22 ± 0.03	0.25 ± 0.02	0.27 ± 0.03	0.33 ± 0.03	0.28 ± 0.06	0.002
Cr (µg/L)	−0.97 ± 0.01	−0.98 ± 0.01	−0.96 ± 0.01	−0.97 ± 0.01	−0.96 ± 0.01	−0.96 ± 0.02	0.432
Mn (µg/L)	0.14 ± 0.02	0.13 ± 0.02	0.11 ± 0.02	0.15 ± 0.02	0.14 ± 0.03	0.17 ± 0.05	0.579
As (µg/L)	1.84 ± 0.02	1.91 ± 0.02 *	1.91 ± 0.01 *	1.95 ± 0.02 *	1.96 ± 0.02 *	1.95 ± 0.04	<0.001
Cu (µg/dL)	0.09 ± 0.01	0.11 ± 0.01	0.13 ± 0.01 *	0.17 ± 0.01 *^†^	0.21 ± 0.01 *^†#^	0.25 ± 0.03 *^†#&^	<0.001

Abbreviations are the same as in Table 1. The data are shown as mean ± standard error of the mean for each log heavy metal. * *p* < 0.05 compared with 0 component; ^†^
*p* < 0.05 compared with 1 component; ^#^
*p* < 0.05 compared with 2 components; ^&^
*p* < 0.05 compared with 3 components.

**Table 4 nutrients-12-02666-t004:** Association of heavy metals with different indices using multivariable linear regression analysis.

Indices	Multivariable	
	Unstandardized Coefficient β (95% CI)	*p*
LAP *		
Urine		
Ni (log per 1 µg/L)	2.418 (0.288, 4.548)	0.026
Cu (log per 1 µg/dL)	9.508 (4.406, 14.609)	<0.001
BRI ^†^		
Blood		
Pb (log per 1 µg/dL)	0.190 (0.019, 0.362)	0.030
Urine		
Cu (log per 1 µg/dL)	0.223 (0.038, 0.407)	0.018
CI ^†^		
Urine		
Cu (log per 1 µg/dL)	0.014 (0.002, 0.027)	0.023
VAI *		
Urine		
Cu (log per 1 µg/dL)	0.898 (0.149, 1.646)	0.019
BAI ^†^		
Blood		
Pb (log per 1 µg/dL)	1.093 (0.241, 1.944)	0.012
AVI ^†^		
Blood		
Pb (log per 1 µg/dL)	0.726 (0.120, 1.332)	0.019

Values are expressed as unstandardized coefficient β and 95% confidence interval (CI). Abbreviations are the same as in Table 1. * Covariates in the multivariable model included age, sex, total cholesterol, hemoglobin, eGFR, and uric acid. ^†^ Covariates in the multivariable model included age, sex, log triglyceride, and total cholesterol.

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
