# Peer review of "Associations of Heavy Metals with Metabolic Syndrome and Anthropometric Indices"

_nutrients, 2020, doi:10.3390/nu12092666_

Round 1

Reviewer 1 Report

The analysis is done rather well. However, there is a potential problem with inflated Type I error--a Bonferroni correction should be applied, to accentuate the really strong results.

The authors should address any difficulties associated with the voluntary/convenience nature of the sample. At a minimum, this makes it considerably more difficult to interpret standard errors and confidence intervals.

Suggestions for policy and practice would be very welcome, and could easily be added to the discussion.

A minor point is that the title might be a bit easier to parse as: "Associations of Heavy Metals with Metabolic Syndrome and Anthropometric Indices." The additional "and" in the original title induces ambiguity otherwise.

Author Response

Reviewer 1

  1. The analysis is done rather well. However, there is a potential problem with inflated Type I error--a Bonferroni correction should be applied, to accentuate the really strong results.*The multiple testing-adjusted corrections p values were computed by stepdown Bonferroni (threshold of p = 0.0083; p0/N1, p0=0.05, N1=6 risk factors, see Table 3). (Line 187-189)Ans: Thank you for raising your concern. We totally agreed your point about inflated type I error. We applied a Bonferroni correction on Table 3.
  2. The authors should address any difficulties associated with the voluntary/convenience nature of the sample. At a minimum, this makes it considerably more difficult to interpret standard errors and confidence intervals. * Fourth, because participants were willing to attend the study were included, which makes it considerably more difficult to interpret standard errors and confidence intervals. (Line 323-325)Ans: Thank you for your comments. We have added this issue in the Limitation.
  3. Suggestions for policy and practice would be very welcome, and could easily be added to the discussion.* MetS per se or its compartments was related to an important public health threat and economic burden by further developing diabetes or cardiovascular events [65,66], and the early diagnosis, follow-up, and long-term management were recommended [67]. Therefore, government might not only make policy to prevent future heavy metals pollution but also continuously monitor the exposers of the specific heavy metals which we have revealed the possible association with MetS in this article. (Line 313-318)Ans: Thank you for your suggestions. We have added the suggestions in the Discussion.
  4. A minor point is that the title might be a bit easier to parse as: "Associations of Heavy Metals with Metabolic Syndrome and Anthropometric Indices." The additional "and" in the original title induces ambiguity otherwise. Ans: Thank you for your corrections. We have revised the title to “"Associations of Heavy Metals with Metabolic Syndrome and Anthropometric Indices". (Line 2-3)

Reviewer 2 Report

The manuscript by Wen et al examined the association between heavy metals and the determinants of the metabolic syndrome. This is an interesting topic with multiple implications. However, there are problems with the statistical analyses and the discussion.

Comments:

  1. The introduction provides a good overview, but the rationale for the study is unclear.
  2. Lines 130: multiple comparisons are conducted after finding main effects. It’s unclear what this is referring to.
  3. Linear regression is not indicated in the stats section.
  4. Table 3 is confusing. It is unclear how the sum of Mets components is examined. For instance, does 1 indicate having 1 component? Also, are medians and IQR presented? If so, an ANOVA cannot be performed on non-normal data.
  5. line 16 states that there is a significant trend for stepwise increases in Cu. What stats were performed here? Also, Pb has a significant p value in the table, but is not mentioned to be significant.
  6. The discussion is formulaic in how it’s written and uses repetitive phrasing. The implications of the findings are not presented.

Author Response

Reviewer 2

  1. The introduction provides a good overview, but the rationale for the study is unclear. Ans: Thank you for your comments. We have added this issue in the Introduction.

* The underlying mechanism of heavy metals toxicity were complex but a large part of it could be attributed to cell apoptosis or carcinogenesis after exhaust of native cellular antioxidants and fail to balance heavy metals-induced reactive oxygen species (ROS) production anymore [22]. This oxidative stress might also decrease the insulin gene promoter activity and mRNA expression in pancreatic islet cells; besides, some of the toxic metals are implicated to disrupt the glucose uptake and alter the related molecular mechanism in glucose regulation directly [23]. All of the above could be dedicated to the relationship between heavy metals and metabolic syndrome. (Line 80-86)

  1. Lines 130: multiple comparisons are conducted after finding main effects. It’s unclear what this is referring to.

Ans: Multiple comparisons among study groups were performed using one-way analysis of variance followed by Bonferroni-adjusted post hoc analysis, which was used for Table 3.

  1. Linear regression is not indicated in the stats section.

Ans: Thank you for your remind. We have added in the Statistical analysis.

* Multivariable linear regression analysis was used to identify associations between heavy metals and LAP, VAI and anthropometric indices. (Line 154-155)

  1. Table 3 is confusing. It is unclear how the sum of Mets components is examined. For instance, does 1 indicate having 1 component? Also, are medians and IQR presented? If so, an ANOVA cannot be performed on non-normal data.

Ans: Sorry for the misunderstanding. Yes, 1 indicated having 1 component. For all heavy metal measurements (in blood and urine), the natural logarithm was used. Therefore, we have changed the presented form as mean ± SE for log heavy metals instead of median (25th -75th percentile).

  1. line 16 states that there is a significant trend for stepwise increases in Cu. What stats were performed here? Also, Pb has a significant p value in the table, but is not mentioned to be significant.

Ans: Thank you for your corrections. We have changed the linear p for trend instead overall p in table 3, and we corrected our sentences. Besides, we also performed Bonferroni-adjusted post hoc analysis of Table 3, we have added in the table 3.

* Table 3 shows heavy metal values according to the sum of MetS components in the participants. The concentration of blood Pb, urine Ni, As and Cu increased with the number of MetS components (from 0 to 5, A linear p ≦ 0.002 for trend). (Line 185-187)

6. The discussion is formulaic in how it’s written and uses repetitive phrasing. The implications of the findings are not presented

Ans: Thank you for your concern. Before manuscript submitting, we have sent for English editing. We have added the implications in the Discussion.

* MetS per se or its compartments was related to an important public health threat and economic burden by further developing diabetes or cardiovascular events [65,66], and the early diagnosis, follow-up, and long-term management were recommended [67]. Therefore, government might not only make policy to prevent future heavy metals pollution but also continuously monitor the exposers of the specific heavy metals which we have revealed the possible association with MetS in this article. (Line 313-318

Reviewer 3 Report

MS: Nutrients-907040

Title: Associations between Heavy Metals and Metabolic 2 Syndrome and Anthropometric Indices

Comments for authors

The authors evaluated the relationships between urine nickel (Ni), copper (Cu), etc. and blood lead (Pb), levels and MetS and different anthropometric indices in moderate (n=2000) sample size. This work provided the consolidated data on heavy metals of concerns (As, Pb, Cu, Ni, etc.) for their role in the emerging issue of metabolic disease.

However, I am not very sure if authors could grasp most of the recent/available epidemiological knowledge and updated accordingly to justify/interpret the presented results. In addition, I have a few concerns to be addressed by authors before any recommendations.

Abstract

  1. …were enrolled . MetS was defined…”, Please, be careful in minor typo like space before “Period”.

Introduction

  1. Well written introduction; short and focused, enjoyed reading it. However, not sure if we can call lipid accumulation product (LAP) under anthropometric indexes as we measure triglycerides (TGs), and high-density lipoprotein (HDL) cholesterol in the blood.
  2. “Previous studies have reported associations 73 between MetS and blood levels of lead (Pb), serum levels of manganese (Mn), chromium (Cr), and 74 selenium (Se), and urine levels of copper (Cu) and zinc (Zn) [20-23].” This statement would be helpful if specified with the type of association by different parameters with different studies.
  3. “However, no study has focused on the relationships between anthropometric indices and heavy metals.” A few more literature to lead such a strong statement/conclusion would be helpful to readers. Here, I am not very sure if authors could grasp most of the recent/available epidemiological knowledge to define the knowledge gap.

Result

  1. Nicely expressed except alignment of variables in Table 2-4, better to do left, i.e., like that of Table 1

Discussion

Well written Discussion; short and focused, enjoyed reading it but it should be elaborated.

  1. The authors did not discuss the Cu, Ni, & other metal exposure levels in the reported study population to locate the exposure level of the population with other populations to interpret the dose of exposure. Further, authors could speculate/propose a potential source of such exposure for further research.
  2. Discussion regarding C-reactive protein comparing other study populations would benefit readers.
  3. The author could speculate potential pathways of MetS considering lots of associations with different anthropometric indicators.
  4. Vehicle emission of lead (Pb) in the air usually considered as one of the sources of Pb exposure but the authors did not assume it in this study. Can we assume zero Pb emission with 100% compliance?
  5. The authors did not cite ref for Korean study regarding Pb and MetS.

Overall, if authors could first discuss exposure level of 3 metals of concern (Cu, Ni, and Pb) to locate their exposure level, discussion regarding association would be easier for the reader to understand the context with other similar studies reporting similar associations. I am not very sure if authors could grasp most of the recent/available epidemiological knowledge and updated accordingly to justify/interpret the presented results. Further, there are lots of other variables associated were not discussed.

I hope, the authors will address these concerns to give its message more effectively.

Author Response

Reviewer 3

The authors evaluated the relationships between urine nickel (Ni), copper (Cu), etc. and blood lead (Pb), levels and MetS and different anthropometric indices in moderate (n=2000) sample size. This work provided the consolidated data on heavy metals of concerns (As, Pb, Cu, Ni, etc.) for their role in the emerging issue of metabolic disease.

However, I am not very sure if authors could grasp most of the recent/available epidemiological knowledge and updated accordingly to justify/interpret the presented results. In addition, I have a few concerns to be addressed by authors before any recommendations.

Abstract

  1. …were enrolled . MetS was defined…”, Please, be careful in minor typo like space before “Period”.

Ans: Thank you for your remind. We have corrected.

Introduction

  1. Well written introduction; short and focused, enjoyed reading it. However, not sure if we can call lipid accumulation product (LAP) under anthropometric indexes as we measure triglycerides (TGs), and high-density lipoprotein (HDL) cholesterol in the blood.

Ans: Thank you for your comments. We totally agreed your point. We will separate them.

*This study aimed to investigate relationships between heavy metals and metabolic syndrome (MetS), lipid accumulation product (LAP), visceral adiposity index (VAI), and anthropometric indices including body roundness index (BRI), conicity index (CI), body adiposity index (BAI) and abdominal volume index (AVI). (Line 26-29)

*These indices include lipid accumulation product (LAP), visceral adiposity index (VAI), and anthropometric indices including body roundness index (BRI), conicity index (CI), body adiposity index (BAI) and abdominal volume index (AVI), all of which can easily be calculated and quantified using factors such as waist circumference, hip circumference, body mass index (BMI), body height (BH), body weight (BW), triglycerides (TGs), and high-density lipoprotein (HDL) cholesterol. (Line 74-79)

  1. “Previous studies have reported associations 73 between MetS and blood levels of lead (Pb), serum levels of manganese (Mn), chromium (Cr), and 74 selenium (Se), and urine levels of copper (Cu) and zinc (Zn) [20-23].” This statement would be helpful if specified with the type of association by different parameters with different studies.

Ans: Thank you for your remind. We have changed the statement.

* Previous studies have reported associations between MetS and several kinds of heavy metals. Higher prevalence of MetS was associated with higher blood levels of lead (Pb) [24] or higher urinary levels of copper (Cu) and zinc (Zn) [25]. Furthermore, a study disclosed significantly higher serum levels of manganese (Mn) and chromium (Cr) in obese aged men and might indirectly intensify MetS, but lower serum levels of magnesium (Mg) increased risk of MetS directly in the same population [26]. (Line 86-91)

  1. “However, no study has focused on the relationships between anthropometric indices and heavy metals.” A few more literature to lead such a strong statement/conclusion would be helpful to readers. Here, I am not very sure if authors could grasp most of the recent/available epidemiological knowledge to define the knowledge gap.

Ans: Thanks for your remind. We have added in the Introduction.

* There were also some studies focused on the association between heavy metals and body composition but more inconsistent. Chromium (Cr) supplementation was associated with mild body weight loss and decrease of body fat percentage in obese or overweight participants [27]. Serum Zn concentrations were positively associated with both abdominal obesity and total body fat in men but not in women [28]. Neither blood Pb concentrations nor urine arsenic (As) concentrations were associated with fat mass percentage in reproductive-age women [29]. (Line 92-97)

Result

  1. Nicely expressed except alignment of variables in Table 2-4, better to do left, i.e., like that of Table 1

Ans: Thank you for your suggestion. We have corrected.

Discussion

Well written Discussion; short and focused, enjoyed reading it but it should be elaborated.

  1. The authors did not discuss the Cu, Ni, & other metal exposure levels in the reported study population to locate the exposure level of the population with other populations to interpret the dose of exposure. Further, authors could speculate/propose a potential source of such exposure for further research.

Ans: Thank you for your comments. We have added the discussion about the exposure level of the population with other populations.

*There was no association between urine Cr or Mn and Mets or any anthropometric index in our study. Chromium (III) or trivalent chromium is an essential trace element that plays a role in carbohydrate and lipid metabolism in the human body and are normally present in blood and urine, with the mean level 0.10–0.16 and 0.22 μg/L respectively [58]. As mentioned before, Cr supplement could decrease BW and improve anthropometric indices in the obese but not already DM population [27]. Chronic exposure to Cr was found to decrease MetS development in United States but not in Korean population [59,60]. The reason why our study did not reveal similar relationship between concentrations and MetS or anthropometric indices may due to generally lower Cr exposure level similar to the Korean study. Mn is also an essential trace element, and the mean urinary manganese concentration in general population was about 1.19 μg/L [61]. Unlike Cr, its deficiency is rarely reported and it is still inconclusive whether low Mn diet in human is related to MetS or not [62,63]. In contrast, Mn poisoning and its neurotoxicity was well documented possibly due to neurons’ long lifespan and high energy demand. Though Mn also accumulate in pancreas, no definite evidence of its toxicity related to MetS was reported, similar to our study [64]. (Line 299-312)

  1. Discussion regarding C-reactive protein comparing other study populations would benefit readers.

Ans: Thank you for your suggestion. However, we have no C-reactive protein data in the present study. We have added the sentence in the Discussion.

*However, in the present study, we did not have the data of C-reactive protein. We could not compare the data with other studies. (Line 244-245)

  1. The author could speculate potential pathways of MetS considering lots of associations with different anthropometric indicators.
    Ans: Thank you for your suggestion. We have added it in the Introduction.

* The potential pathways of these central obesity-representing indices associated with MetS could be related to visceral fat increasing production of nonesterified fatty acids (NEFA), inflammatory cytokines, prothrombotic factor such as plasminogen activator inhibitor (PAI-1), and activation of renin-angiotensin-aldosterone (RAAS) system, but decrease production of an anti-inflammatory and antiatherogenic adipokine, adiponectin [20,21]. (Line 70-74)

  1. Vehicle emission of lead (Pb) in the air usually considered as one of the sources of Pb exposure but the authors did not assume it in this study. Can we assume zero Pb emission with 100% compliance?

Ans: Thank you for your comments. We have added the source of vehicle emission, and added the issue about the source in the limitation.
* Lead is another common pollutant, and the most likely exposure routes in the general population are ingestion of contaminated food and drinking water, possibly through water transmitted by lead pipes or food contained in improperly glazed pottery dishes; furthermore, vehicle emission of lead in the air is also one of the sources of Pb exposure in area still using leaded gasoline [51]. (Line 263-267)

* In addition, the source of these heavy metals cannot be confirmed. (Line 325-326)

  1. The authors did not cite ref for Korean study regarding Pb and MetS.

Overall, if authors could first discuss exposure level of 3 metals of concern (Cu, Ni, and Pb) to locate their exposure level, discussion regarding association would be easier for the reader to understand the context with other similar studies reporting similar associations. I am not very sure if authors could grasp most of the recent/available epidemiological knowledge and updated accordingly to justify/interpret the presented results. Further, there are lots of other variables associated were not discussed.

I hope, the authors will address these concerns to give its message more effectively

Ans: Thank you for your comments. We have cited the reference, and tried to grasp most of the recent/available epidemiological knowledge and updated. We have added some more papers to discuss. We have also added exposure level comparison to other similar studies.

* Chronic arsenic exposure threatens more than 100 million people’s health worldwide where drinking water sources were groundwater with higher concentration of this toxic metalloid, including certain areas of Taiwan [54]. Besides blackfoot disease or cancers (skin, bladder, lung) which were increasingly found in high-dose long term exposure, increased risk of MetS was also reported. Wang et all have reported an increased prevalence of MetS from the 2nd tertile of hair As concentrations in population at central part of Taiwan, but the urine As concentration data was not available [55]. In the present study, though we didn’t find a direct relationship between urine As levels and metabolic syndrome, we did find urine As trend was accompanied with an increasing number of MetS components. The discrepancy might be attributed to no measurement of the composition of total urine As, which could be divided into inorganic arsenic (iAs), dimethylarsinic acid (DMA), and monomethylarsonic acid (MMA). The latter two organic As ratio represented individualized arsenic methylation capability, which was suspected to be a more important factor of metabolic risk than merely total urine As concentration in no matter high or low to moderately As-exposed population (urine total As: 6.5- ~43.46 μg/L) [56,57].

There was no association between urine Cr or Mn and Mets or any anthropometric index in our study. Chromium (III) or trivalent chromium is an essential trace element that plays a role in carbohydrate and lipid metabolism in the human body and are normally present in blood and urine, with the mean level 0.10–0.16 and 0.22 μg/L respectively [58]. As mentioned before, Cr supplement could decrease BW and improve anthropometric indices in the obese but not already DM population [27]. Chronic exposure to Cr was found to decrease MetS development in United States but not in Korean population [59,60]. The reason why our study did not reveal similar relationship between concentrations and MetS or anthropometric indices may due to generally lower Cr exposure level similar to the Korean study. Mn is also an essential trace element, and the mean urinary manganese concentration in general population was about 1.19 μg/L [61]. Unlike Cr, its deficiency is rarely reported and it is still inconclusive whether low Mn diet in human is related to MetS or not [62,63]. In contrast, Mn poisoning and its neurotoxicity was well documented possibly due to neurons’ long lifespan and high energy demand. Though Mn also accumulate in pancreas, no definite evidence of its toxicity related to MetS was reported, similar to our study [64]. (Line 285-312)

Round 2

Reviewer 1 Report

Other than editing this seems ready for publication.

Author Response

Other than editing this seems ready for publication

Ans: Thank you very much.

Reviewer 2 Report

I am satisfied with the statistical changes, but the discussion needs editing (for instance, in every paragraph, this sentence is repeated "however few studies have investigated the effect on metabolic diseases...", as well as others, which make the discussion hard to read. 

There are also grammatical changes that need to be corrected in the new text as well. 

Author Response

I am satisfied with the statistical changes, but the discussion needs editing (for instance, in every paragraph, this sentence is repeated "however few studies have investigated the effect on metabolic diseases...", as well as others, which make the discussion hard to read.

Ans: Thank you for your comments. We have revised our sentences.

There are also grammatical changes that need to be corrected in the new text as well.

Ans: Thank you for your comments. We have corrected some grammatical changes.

Reviewer 3 Report

MS: Nutrients-907040R1

Title: Associations of Heavy Metals with Metabolic Syndrome and Anthropometric Indices

Comments for authors

I commend the authors on their responsivity to reviewers' comments and overall, they adequately addressed my concerns.  

Hence, I recommend for publication of this manuscript with some edits, if any listed below.

  1. .. “To be additional”, replace by “In addition”, Page 8 Line 254
  2. “…several kinds of heavy metals” replace by “different heavy metals” Page 2, Line 87
  3. In Table 3, Indent set as done as in Table 2 and 4 will be nice.
  4. Wang et all” should be “Wang et al.” [?] Page 8, Line 291
  5. ..”anthropometric indices may due to..” should be “anthropometric indices may be due to” Page 9, Line 308.

Author Response

I commend the authors on their responsivity to reviewers' comments and overall, they adequately addressed my concerns.

Hence, I recommend for publication of this manuscript with some edits, if any listed below.

.. “To be additional”, replace by “In addition”, Page 8 Line 254

“…several kinds of heavy metals” replace by “different heavy metals” Page 2, Line 87

In Table 3, Indent set as done as in Table 2 and 4 will be nice.

“Wang et all” should be “Wang et al.” [?] Page 8, Line 291

..”anthropometric indices may due to..” should be “anthropometric indices may be due to” Page 9, Line 308.

Ans: Thank you for your corrections. We have corrected these sentences and table.
